



# 25 years of Cloud Base Height Measurements by Ceilometer in Ny-Ålesund, Svalbard

Marion Maturilli[1], Kerstin Ebell[2]

[1] Alfred Wegener Institute Helmholtz Centre for Polar and Marine Research, Potsdam, D-14473 Germany
Institute for Geophysics and Meteorology, University of Cologne, D-50969 Cologne, Germany

*Correspondence to*: Marion Maturilli (marion.maturilli@awi.de)

**Abstract.** Clouds are a key factor for the Arctic Amplification of global warming, but their actual appearance and distribution is still afflicted with large uncertainty. On the Arctic wide scale, large discrepancies are found between the various reanalyses and satellite products, respectively. Although ground-based observations by remote sensing are limited to point measurements, they have the advantage to obtain extended time series of vertically resolved cloud properties. Here, we present a 25-year data record of cloud base height measured by ceilometer at the Arctic site Ny-Ålesund, Svalbard. Linked to cyclonic activity, the cloud base height provides essential information for the interpretation of the surface radiation balance and contributes to the understanding meteorological processes. Furthermore, it is a useful auxiliary component for the analysis of advanced technologies that provide insight to cloud microphysical properties, like the cloud radar. The long-term time series also allows deriving an annual cycle of the cloud occurrence frequency, revealing the more frequent cloud cover in summer and the lowest cloud cover amount in April. However, as the use of different ceilometer instruments over the years potentially imposed inhomogeneities to the data record, any long-term trend analysis should be avoided.

The Ny-Ålesund cloud base height data from August 1992 to July 2017 are provided in high temporal resolution of 5 minutes (1 minute) before (after) July 1998, respectively, at the PANGAEA repository (doi:10.1594/PANGAEA.880300).

## 1 Introduction

The Arctic amplification of climate warming is attributed to several factors and their feedback processes in the climate system. Contributing to the enhanced high-latitude warming are the sea ice – albedo effect, the lapse rate feedback (Pithan and Mauritsen, 2015), atmospheric heat and moisture advection (Park et al, 2015), ocean heat transport (Chylek et al., 2009), aerosol effects and potentially others, all linked in complex relations. Particularly clouds are known as a major contributor to Arctic amplification (Curry et al., 1996). As a result of the warmer and moister climate, both macrophysical (e.g. cloud base height, vertical geometric thickness, horizontal extent) and microphysical (e.g. particle size and phase) characteristics of Arctic clouds may change, affecting the radiation budget in this sensible part of the climate system. Especially low-level mixed-phase clouds have the potential to significantly contribute to Arctic warming (Bennartz et al., 2013), while their formation and persistence is not well captured in numerical models. The uncertain range of Arctic clouds and their radiative impact contribute to the large spread across current climate models (Karlsson and Svensson, 2011).





Moreover, Arctic wide observations of clouds by satellite show discrepancies between data sets that arise from differences in instruments and cloud detection algorithms which are crucial under typical Arctic conditions with very low thermal and radiance contrasts between clouds and the underlying ice and snow surface (Chernokulsky and Mokhov, 2012). Surface based remote sensing is limited to point observations that potentially are not representative of a larger area, but have the advantage

of higher vertical and temporal resolution. Though the combination of more sophisticated techniques provides detailed observational data of various cloud parameters (Shupe et al., 2008), even the perspicuous and easy-to-retrieve cloud base height data by ceilometer measurements contribute to a better understanding of the general properties of Arctic clouds. Under cloudy conditions, the cloud base is the effective height of downward long-wave emission, and thus affects the downward component of long-wave radiation observed at the surface. Here, we present a 25-year ceilometer cloud base height dataset

from Ny-Ålesund, Svalbard, and indicate the potential application areas.

## 2 Data

Among the broad suite of instrumentation in Ny-Ålesund, Svalbard, the Alfred Wegener Institute operates laser ceilometer measurements for the detection of cloud base height since August 1992 (Maturilli and Herber, 2017). The measurement principle is that of light detection and ranging (lidar): a pulsed laser beam is sent vertically to the atmosphere, where light is

scattered back by air molecules and particles. By detecting the run-time of the return signal, the ceilometer identifies the lowest altitude of a cloud as layer with higher particle backscatter characteristics. For the more powerful systems it is possible to detect up to three cloud layers if the lower cloud deck is optically thin enough to allow the transmittance of the laser beam and the backscattered light from the upper cloud layer. The described Ny-Ålesund data have been retrieved by commercial ceilometer systems, and the corresponding operational software was used to identify the cloud base height (CBH). In the

presented period August 1992 to July 2017, three different instruments have been applied for the measurements: (1) LD-WHX by Impulsphysik GmbH, (2) LD-40 by Vaisala, and (3) CL-51 by Vaisala (Figure 1, respectively). Missing months in the data series are February 1993, February to May 1997, December 1999 to March 2000, and March 2000, due to technical problems with the instrumentation. In all available months, the code 99999 marks that no cloud has been detected. Details on the associated measurement periods of the respective instruments and the data resolution are given in Table 1.

Obviously, the instruments have different upper detection limits for the cloud base height. While the older instrument did not report clouds above 3650 m and was thus blind for high clouds, the newer instruments easily cover the whole troposphere. Furthermore, it is likely that higher laser power and more sensitive receiving hardware increased the sensitivity for cloud detection in the newer systems, potentially affecting the observed frequency of clear sky conditions. Overall, every renewal of the instrumentation brought an increase in sensitivity and precision in cloud base detection. This qualitative improvement has

positive effects on the recent analysis of cloud related process studies. On the other hand, the change in instrumentation leads to inhomogeneity of the dataset regarding the long-term climate record. Both aspects will be highlighted in the next sections.



## 3 Cloud Base Height for Process Studies

The high temporal resolution of 1 minute (5 minutes before August 1998, respectively) of the cloud base height data enables detailed process studies of the changing cloud structure and its relation to varying meteorological conditions.

### 3.1 Cloud Base Height in Meteorological Context

Clouds are an indicator of the synoptic situation, and the cloud base height provided by the ceilometer is associated with e.g. the changing cloud deck during the passage of a frontal system. Here, we show an example of a small cyclonic system passing Ny-Ålesund on 15 / 16 December 2016 (Figure 2). In the early morning hours of 15 December, temperatures at the surface were a few degrees below freezing, with a compact cloud deck with base heights in about 400 to 700 m. The clouds disappeared at about 10:40 UTC, unveiling clear sky conditions for about one hour. At 11:30 UTC, clouds appear with a base height at 6

km, descending during the next 7 hours to cloud base heights below 1 km. The descent of the cloud base is a typical feature of an approaching warm front that is inclined forward in the upper atmosphere as indicated in the schematic diagram in the upper panel of Figure 2. Indeed, during this warm front phase an increase of temperature is observed from about -4°C to about 0°C. Since by this time the effect of warm air advection within the cyclone's warm air sector has not yet reached the surface, the increase in 2 m air temperature is likely related to the radiative effect of the warm front associated cloud cover. As the

ceilometer is operated in close vicinity to the instrumental set-up for the Baseline Surface Radiation Network (BSRN), all surface radiation balance parameters are available. The presented case refers to December, implying polar night conditions at Ny-Ålesund and therefore reducing the contribution to the radiation balance to the long-wave components. Basically, the upward long-wave radiation $LW_{up}$ closely follows the air temperature while the downward long-wave radiation $LW_{down}$ is very much affected by the presence of clouds in the atmospheric column. They both contribute to the surface net long-wave radiation

$LW_{net}$ which is close to zero for overcast conditions and can take large negative values for clear sky conditions, also known as the 'cloudy' and 'opaque' Arctic winter states (Stramler et al., 2011), respectively. Indeed we find a large difference for the up- and downward long-wave radiation of about $LW_{net} = LW_{up} - LW_{down} = -60 \, Wm^{-2}$ during the clear sky hour before the first cirrus clouds of the warm front occur. This difference reduces as the cloud base descents. The resulting increase in $LW_{net}$ contributes to the observed increase in temperature. By about 19 UTC, the warm front also passes at surface level, accompanied

by an intensification of surface wind speed (not shown). The now present warm air sector of the cyclone is associated with low stratiform clouds, representing the 'opaque' state with net long-wave radiation around 0 $Wm^{-2}$. The temperature is stably warm around freezing until the cloud deck loosens up in the early morning hours of 16 December. Once the clouds get patchier, the downward long-wave radiation partly arrives from higher and colder parts of the atmosphere, resulting in an interplay of 'clear' and 'opaque' radiation conditions and according air temperature fluctuations. In these conditions, also the ceilometer

laser beam can pass in between the lowermost clouds, receiving the cloud base height from an upper level cloud. The cyclonic influence ends by about 15:30 UTC, when the clouds disappear, the long-wave radiation drops back to the 'clear' state, and air temperature decreases. It remains unclear if these changes are caused by a weak cold front, as neither a change in wind



speed nor wind direction were observed, and the remnants of the cyclone dissolved over Svalbard before the next cyclone approached.

Overall, the cloud base height provided by the ceilometer allows an interpretation of the involved synoptic cloud types, and an estimation of their effect on the surface radiation.

**3.2 Cloud Base Height as Auxiliary for In-Situ and Remote Sensing Cloud Measurements**

The presence of clouds and their radiative effects in the Arctic are crucial topics to the understanding of Arctic climate change. To approach the comprehensive characterization of macro- and micrpophysical cloud parameters in Ny-Ålesund, a 94 GHz frequency modulated continuous wave cloud radar (Küchler et al., 2017) has been installed on the roof of the AWIPEV atmospheric observatory in June 2016 by the University of Cologne within the frame of the Transregional Collaborative Research Center (TR 172) "ArctiC Amplification: Climate Relevant Atmospheric and SurfaCe Processes, and Feedback Mechanisms (AC)[3]" (Wendisch et al., 2017). The cloud radar provides vertical profiles of radar reflectivity factor, Doppler velocity, and Doppler spectral width from 150 m to 10 km above ground. An example of a time-height series of the radar reflectivity factor, which has been measured at the AWIPEV atmospheric observatory on 23 Nov 2016 between 10 to 20 UTC, is given in Figure 3. In contrast to a lidar instrument, which is very sensitive to small particles like cloud droplets and aerosol, a cloud radar is also sensitive to larger particles such as rain, drizzle drops and snow. The observed backscattered signal of the cloud radar might thus be generated by different hydrometeor types. Therefore, it is very difficult to discriminate between liquid cloud droplets and precipitating particles (rain or snow) from cloud radar observations alone. By including ceilometer observations, e.g. the ceilometer cloud base height (black dots in Figure 3), we can better identify cloud droplet layers, i.e. layers where the lidar backscatter is high. On 23 Nov 2016, such a liquid layer is observed by the ceilometer with cloud base heights at ~1 km at 10 UTC to 1.4 km at 18 UTC. Below the observed cloud base height, the radar signal is caused by precipitating particles. Based on vertical temperature information, we even know that the liquid layer is supercooled and that the precipitation below is snow. The cloud observed on 23 Nov 2016 is a typical example of an Arctic low-level mixed-phase cloud. Only the combination of cloud radar and ceilometer thus allows for a comprehensive view on this ubiquitous Arctic cloud type.

**4 Cloud Base Height for Long-Term Climate Studies**

With cloud base measurements available since 1992 and temperature and radiation changes observed during the same period (Maturilli et al., 2013; Maturilli et al., 2015), it seems a natural consequence to analyse the ceilometer data set with regard to long-term changes in cloud base height. Yet, here we want to emphasize that the data set is not suited for long-term trend analysis due to the inhomogeneity of the data retrieval within the time series. The different instrumentation used over the years (Table 1) had diverse sensitivities and maximum detection limits, and even for a single instrument a drift in sensitivity over



the years cannot be excluded. Therefore, potential changes shown here may not be unambiguously attributed to actual changes in the atmosphere, but may be due to the inhomogeneity of the data caused by the different instrumentation.

The annual cycle of the clear sky and cloudy conditions in Ny-Ålesund is shown in Figure 4. Here, the data have been divided to subsets of different instruments, referring to August 1992 to June 1998 (LD-WHX05), August 1998 to June 2011 (LD-40), and September 2011 to July 2017 (CL-51). The overlapping months July 1998 and August 2011 have been excluded, as well as all months that had more than 20% missing data. Obviously, the early instrument with the lowest cloud height detection limit apparently identified the largest percentage of clear sky cases, with the largest deviation to the other subsets in late autumn and winter. Consequently, less cloud cases were detected in the early period, most likely caused by the instrument's low cloud height detection limit and thus blindness to clouds above 3650 m. Still, differences also occur between the subsets of the later two instruments. If a change in occurrence frequency of clouds over Ny-Ålesund occurred over the 25-year period, it would unfortunately be masked by the different sensitivity of the different ceilometers. Nevertheless, some general conclusions can be drawn from the observations: The extended summer season May to September reveals the lowest occurrence of clear sky conditions, and is consequently the most cloud covered period of the year. This is in line with other Arctic sites where the summer season is characterized by persistent low cloud cover (Shupe et al., 2011). In Ny-Ålesund, this is also the season with surface temperatures above freezing and a partly snow-free surface structure (Maturilli et al., 2013). As the largest agreement between all subsets is in July and August, it is likely that the majority of clouds in these months occur as low cloud deck in the lower part of the atmosphere.

Regarding a potential change of the cloud base height over the 25-year period, Figure 5 shows the observed seasonal median cloud base height. The periods of different instrumentation are indicated, and a shift in CBH from the first to the second instrument becomes apparent in all seasons. The period after July 1998 does not exhibit any obvious jumps related to the change in instrumentation. Overall, no significant changes in CBH are detected over these years. Yet, the presented ceilometer cloud base height dataset is a valuable long-term source for studies relating atmospheric temperature and humidity data as well as surface radiation measurements with generic cloud information.

## 5 Summary

With 25 years of observations of cloud base height by ceilometer in Ny-Ålesund, Svalbard, we present a long-term data set that contributes to the understanding of cloud processes in an Arctic environment. Due to the inhomogeneity caused by different instrumentation over the years, it is impossible to retrieve any trend that can be unambiguously attributed to changes in the atmosphere. We therefore strongly recommend avoiding any trend analysis based on the presented data set. Nevertheless, the ceilometer data provide a useful contribution to synoptic and cloud studies on shorter time scales. In this context, we presented examples on the passage of a frontal system as well as the combination of cloud base height observations with surface radiation measurements. Furthermore, the ceilometer data prove to provide necessary auxiliary information for the retrieval of cloud parameters from the cloud radar.



The described ceilometer data for August 1992 to July 2017 are available at https://doi.pangaea.de/10.1594/PANGAEA.880300. As the measurements are continuing, additional data after July 2017 can be found with the search term "Expanded measurements from station Ny-Ålesund" at the PANGAEA repository.

**Acknowledgment**

This study has been supported by the SFB/TR172 'Arctic Amplification: Climate Relevant Atmospheric and Surface Processes, and Feedback Mechanisms (AC)3' funded by the Deutsche Forschungsgemeinschaft (DFG).

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



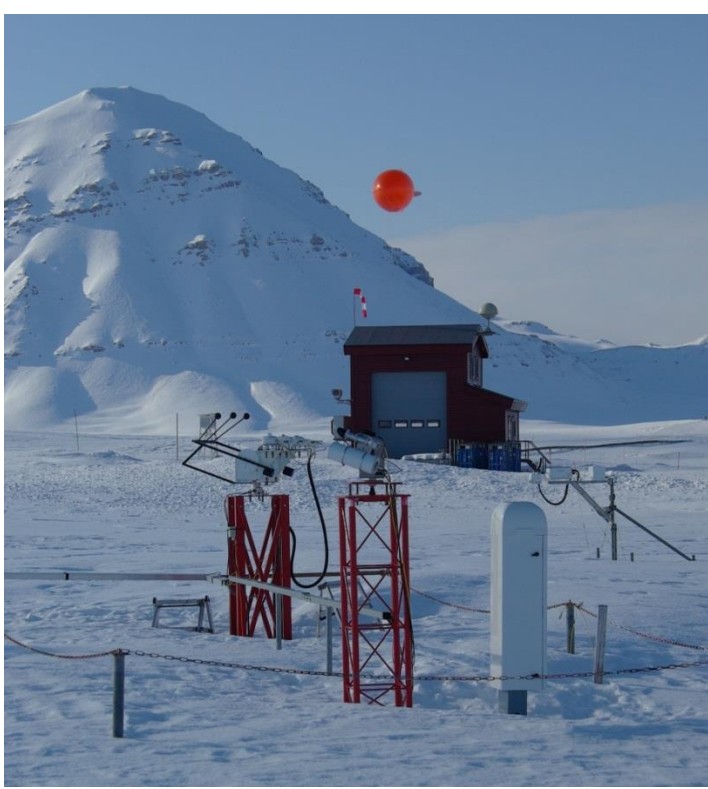

**Figure 1: The CL-51 ceilometer located in the vicinity of the radiation measurements of the AWIPEV station at Ny-Ålesund, in April 2013. In the background the balloon facility for radiosonde launches, and a red tethered balloon. (photo: J. Graeser)**





**Figure 2: A frontal passage on 15 / 16 December 2016 in Ny-Ålesund. a: Schematic diagram of the warm front (red line) and cold front (blue line), their moving direction (black arrow), and associated clouds, respectively. b: Cloud base height (CBH) from ceilometer measurements. c: 2m air temperature from surface meteorological observations. d: Upward and downward longwave radiation (blue and red lines, respectively) from surface radiation measurements.**



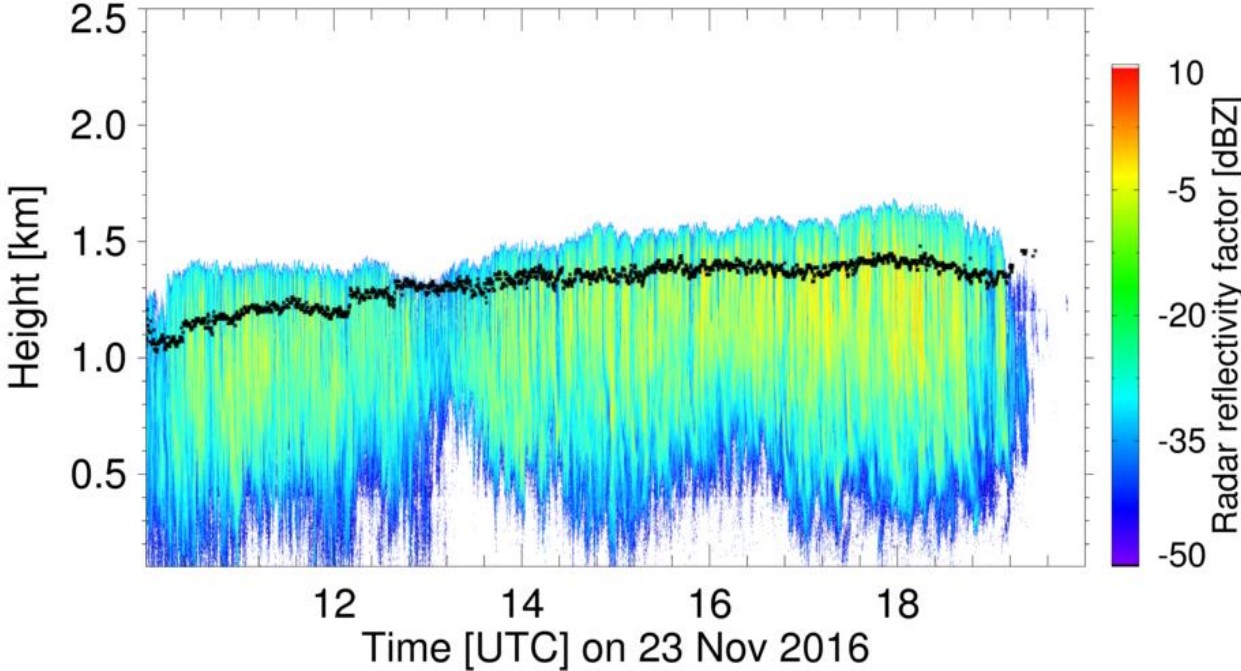

**Figure 3: Time-height series of the cloud radar reflectivity factor on 23 Nov 2016 between 10 and 20 UTC at Ny-Ålesund. Black symbols indicate the cloud base height from ceilometer measurements.**





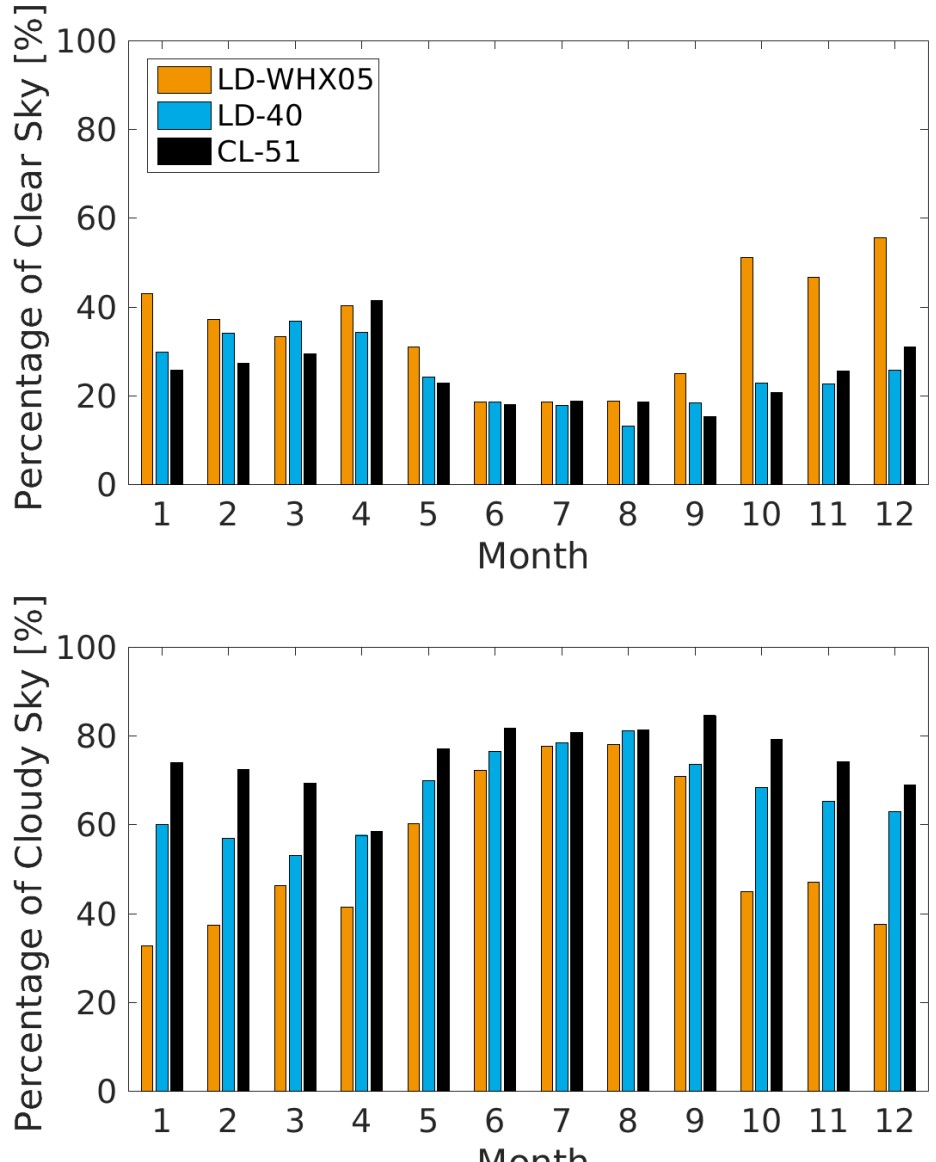

**Figure 4:** Annual cycle of the detected occurrence frequency of clear sky conditions (upper panel) and cloudy conditions (lower panel), given in percent of the total monthly observation times. The average monthly percentage is retrieved from monthly mean values excluding the instrumental overlap months July 1998 and August 2011as well as months with less than 75% data coverage. The 25-year observation period is presented in the subsets August 1992 to June 1998 (orange), August 1998 to June 2011 (blue), and September 2011 to July 2017 (black) according to the different ceilometer types.



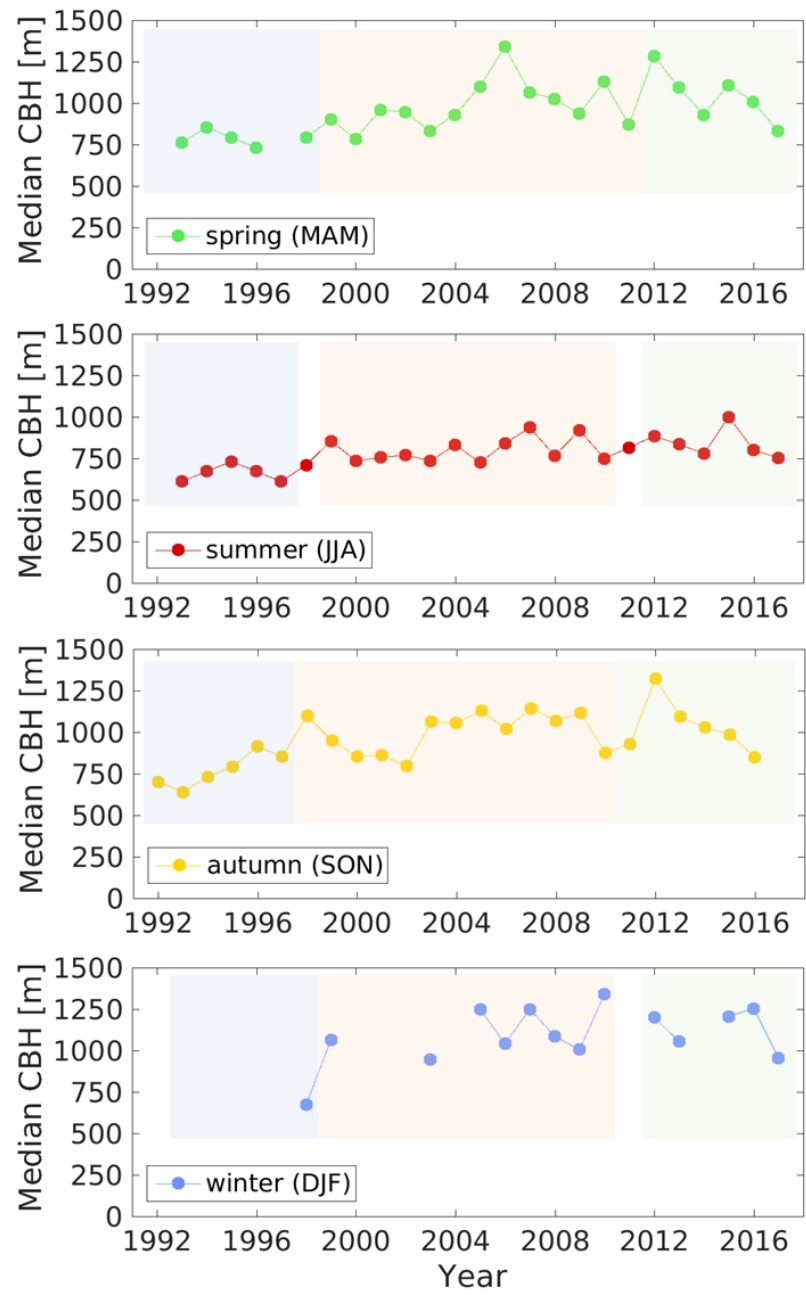

**Figure 5: Median cloud base height (CBH) for spring months March-April-May, summer months June-July-August, autumn months September-October-November, and winter months December-January-February, shown from uppermost to lowermost panel, respectively. Only data with >20% available cloud detection data during the season are considered, leading to gaps especially in winter. The background shading indicates the different instrumentation (light blue – LD-WHX, light red – LD-40, light green – CL-51).**



| Instrument type | **LD-WHX05** | **LD-40** | **CL-51** |
|---|---|---|---|
| Observation period | 1 August 1992 to 13 July 1998 | 14 July 1998 to 24 August 2011 | 25 August 2011 to at least July 2017 |
| provided temporal resolution | 5 min | 1 min | 1 min |
| cloud reporting range | 3650 m | 13000 m | 13000 m |
| vertical resolution | 10 m (below 300m) 20 m (above 320 m) | 7.5 m | 10 m |

5 **Table 1: The three instrument types that contributed to the cloud base height dataset by ceilometer measurements in Ny-Ålesund, and the temporal and spatial resolution of their data.**

