# Peer review of "years of Cloud Base Height Measurements by Ceilometer in Ny-Ålesund, Svalbard"

_Earth System Science Data, 2018_

## Referee Comment (RC1) · Anonymous Referee #1 · 24 Apr 2018

This manuscript presents a good reference for an extense dataset of cloud base height (CBH)measurements by ceilometer. CBH is a very important measure to estimate the downwelling longwave irradiance at the surface. This magnitude is mentioned in the mansucript insection 3.1. It would be interesting include the specifications of the instruments (upwelling and donwelling) used. I would also suggest to include this magnitude averaged for cloudy cases in Figure 5. Addtionally, because there are measurements from satellite (CALIPSO ans CloudSat), I would suggest to compare for the available cases or at least some cases, the CBH measured from the ceilometer to those measured from CALIPSO and CloudSat.

---

## Referee Comment (RC2) · Anonymous Referee #2 · 23 May 2018

This manuscript presents an extensive data set of Ceilometer measurements at Ny-Alesund, Svalbard over 25 years. It explains the characteristics of the data and shows different examples what the data can be useful for. I think it is great to make this data set available to use and suggest accepting the manuscript with minor revisions.

Specific comments:

- I am missing a discussion of the uncertainty range of the presented data set (or each sub data set). It is mentioned at several places that the data can not be used for long-term trend analysis because it is very instrument dependent or different between the different instrument periods. Given that it should be mentioned what the general

uncertainty range is and what the constraints are when using the data.

- The data set presented here consists of three time periods where the data was derived with different instruments. There is no overlap between all three instruments, which makes sense in terms of having one complete consistent data set. However, in case there was an overlap between the different instruments it would be interesting to add an comparison as an aspect referring to the sensitivity/uncertainty of the overall data set.

- Looking at the link how to download the data it seams that the download has to be done on a monthly basis? Please provide one data file for the whole data set presented here or some way to easily download it as a whole since it is also presented as one data set in the manuscript. In case that is available but I was not able to find it, make sure that the website is easy to navigate.

- page 3, line 31: Could there be a microphysical reason for the disappearance of this cloud? Wha there some precipitation or graupel or similar observed?

- page 5, line 6: How often did it happen that a month had more than 20% missing data and was excluded?

- page 5, line 11: You say here that the signal could be masked by the different sensitivities of the different ceilometers, but you don't really discuss how large the uncertainty of the signal and the sensitivities of the different ceilometers are. It would be good to have some information here to relate to.

- Figure 1: Mark which instrument is the ceilometer in the picture.

- Figure 2: The abbreviations of the cloud types might need to be explained.

- Figure 2: Change to a vector graphic.

- Figure 5: The shading is very light and difficult to see. It disappeared on my print-out. Check again to make sure it can be clearly seen.

- Table 1: In the text everywhere it is mentioned that the technology got better, the vertical resolution however went down from 1998-2011 to the actual data set from 2011 ongoing. Please comment this (and change in the text accordingly).

Small remarks,typos:

- page 1, line 22: exchange climate with global or add global to emphasize that this refers to a global mean.

- page 2, line 2: add , before which.

- page 2, line 26: replace was with were.

- page 3, line 9: avoid line break between number and unit.

- page 3, line 26: replace stably by stable.

- caption Fig. 3: replace symbols with dots.

---

## Referee Comment (RC3) · Anonymous Referee #3 · 31 May 2018

Review of the manuscript "25 years of Cloud Base Height Measurements by Ceilometer in Ny-Ålesund, Svalbard" by M. Maturilli and K. Ebell in Earth Syst. Sci. Data Journal (essd-2018-48).

The presented paper uses a 25 year data set of cloud base height measured with three different ceilometers (three different periods). The authors present a description of a case about a small cyclonic system based on CBH data and also longwave irradiance and temperature data. They also add ceilometer data in order to understand better the data provided by a cloud radar. Finally, a statistical analysis about the percentage of cloudy sky is discussed and also the time series of the CBH median, indicating that

results could not be representative since the inhomogeneities in the time series caused by the changes on the instrument. In general this work fits with the scope of the journal and I consider it must be accepted after some minor revisions.

- Abstract: The abstract should not be equal to the introduction section, it should describe a little more what authors will do in the full paper.

- The objective of the paper is not clear in the text.

- Figure 1 does not provide any significant information, it could be removed.

- P4L6: This sentence should be in the introduction but not in this section.

- P4L7: The description of cloud radar fits better in the data section "data".

- P5L18: The median of CBH trends should be quantified by Theil-Sen estimator or others, even homogeneity test could be done in order to stablish the inhomogeneities of the time series. In addition, the time series and trends of the percentage of cloudy sky should add more value to the paper.

- P5L28: but a homogeneity analysis could be done, or at least study the trends in the period of each ceilometer.

- P5L31-32: The results of this work do generally not prove that.

- Figure 4: A legend could help to understand the bars.

---

## Author Comment (AC1) · 11 Jul 2018

**General comment by authors:**

We appreciate the generally positive and encouraging feedback from the three reviewers. We also acknowledge the reviewer comments which led to improvements of the manuscript.

Below, the reviewer's comments are recalled in black, our response is given in blue, and changes to the manuscript are given green.

**Anonymous Referee #1**

This manuscript presents a good reference for an extense dataset of cloud base height (CBH)measurements by ceilometer. CBH is a very important measure to estimate the downwelling longwave irradiance at the surface. This magnitude is mentioned in the mansucript in section 3.1. It would be interesting include the specifications of the instruments (upwelling and donwelling) used.

We agree that this information is important, and added the according information and a reference to the section.

>> As the ceilometer is operated in close vicinity to the instrumental set-up for the Baseline Surface Radiation Network (BSRN) **described in Maturilli et al. (2015)**, all surface radiation balance parameters are available. The presented case refers to December, implying polar night conditions at Ny-Ålesund and therefore reducing the contribution to the radiation balance to the long-wave components **measured by Eppley PIR pyrgeometers**.

I would also suggest to include this magnitude averaged for cloudy cases in Figure 5.

We have added the longwave net radiation for the cloudy cases in Figure 5.

>> page 5, lines 27 pp.
Regarding a potential change of the cloud base height over the 25-year period, Figure 5 shows the observed seasonal median cloud base height, together with the longwave net radiation LWnet = LWdown – LWup from the BSRN surface radiation measurements for simultaneous times. The periods of different ceilometer instrumentation are indicated…

>> Figure caption:
Figure 5: Median cloud base height **(dots; left axis)** for spring months March-April-May, summer months June-July-August, autumn months September-October-November, and winter months December-January-February, shown from uppermost to lowermost panel, respectively. Only data with >20% available cloud detection data during the season are considered, leading to gaps especially in winter. The background shading indicates the different instrumentation (light blue – LD-WHX, light red – LD-40, light yellow – CL-51). **Additionally, the median longwave net radiation LWdown – LWup from simultaneous BSRN surface radiation measurements is shown for the same cloudy periods (triangles; right axis), respectively.**

Addtionally, because there are measurements from satellite (CALIPSO and CloudSat), I would suggest to compare for the available cases or at least some cases, the CBH measured from the ceilometer to those measured from CALIPSO and CloudSat.

Cloudsat and Calipso data have been used to analyze cloud properties in the Arctic (e.g. Mioche et al, 2015). However, cloud base height is a variable which is rather difficult to retrieve from satellite based remote sensing instrumentation due to several reasons. CloudSat provides information on the vertical profile of hydrometeors (cloud particles + precipitation). In the presence of precipitating hydrometeors, the detection of cloud base height from cloud radar measurements alone is thus not possible (as demonstrated for the ground-based radar measurements in Fig. 3). Another serious problem is the presence of the "blind zone" in the CloudSat obervations:  This blind zone is caused by ground-clutter contamination of the CloudSat radar and covers the lowest 1200 m above land/ice surface (Marchand et al., 2008; Maahn et al., 2014). Since low-level clouds with CBHs lower then 1 km are very common in the Arctic (Shupe et al., 2011), CloudSat will always miss these low-level clouds. This will be likely the case for most of the clouds in the case studies presented in this paper having CBHs of ~1 km. This issue is also discussed in more detail in Mioche et al. (2015) who also perfomed a comparison between combined space-borne radar/lidar observations and ground-based lidar observations at Ny-Ålesund. They showed that uncertainties in satellite-based in cloud fraction are 20-25 % between 500 m and 2 km.

As for the ground-based ceilometers, CALIPSO very well detects cloud layers with higher particle backscatter characteristics, in particular liquid layers. In this way, the upper part of the cloud and thus cloud top height can be very well detected, but due to the strong attenuation of the lidar signal, CALIPSO might not even see down to the cloud base.

Due to these well known limitations in the Cloudsat and CALIPSO observations of low-level clouds and the CBH in particular, adding these observations will not provide an added value to the manuscript.

*References*

*Maahn, M., C. Burgard, S. Crewell, I. V. Gorodetskaya, S. Kneifel, S. Lhermitte, K. Van Tricht, andN. P. M. van Lipzig (2014), How does the spaceborne radar blind zone affect derived surface snowfall statis- tics in polar regions?, J. Geophys. Res. Atmos., 119, 13,604–13,620, doi:10.1002/2014JD022079*

*Marchand, R., G. G. Mace, T. Ackerman, and G. Stephens (2008), Hydrometeor detection using CloudSat—An Earth-orbiting 94-GHz cloud radar, J. Atmos. Oceanic Technol., 25(4), 519–533, doi:10.1175/2007JTECHA1006.1*

*Mioche, G., O. Jourdan, M. Ceccaldi, and J. Delanoë (2015), Variability of mixed-phase clouds in the Arctic with a focus on the Svalbard region: a study based on spaceborne active remote sensing, 15, 2445–2461.*

*Shupe, M. D., V. P. Walden, E. Eloranta, T. Uttal, J. R. Campbell, S. M. Starkweather, and M. Shiobara (2011), Clouds at Arctic Atmospheric Observatories. Part I: Occurrence and Macrophysical Properties. Journal of Applied Meteorology and Climatology, 50, 626–644, doi: 10.1175/2010JAMC2467.1*

---

## Author Comment (AC2) · 11 Jul 2018

**General comment by authors:**

We appreciate the generally positive and encouraging feedback from the three reviewers. We also acknowledge the reviewer comments which led to improvements of the manuscript.

Below, the reviewer's comments are recalled in black, our response is given in blue, and changes to the manuscript are given green.

**Anonymous Referee #2**

This manuscript presents an extensive data set of Ceilometer measurements at Ny-Alesund, Svalbard over 25 years. It explains the characteristics of the data and shows different examples what the data can be useful for. I think it is great to make this data set available to use and suggest accepting the manuscript with minor revisions.

Specific comments:

- I am missing a discussion of the uncertainty range of the presented data set (or each sub data set). It is mentioned at several places that the data can not be used for long-term trend analysis because it is very instrument dependent or different between the different instrument periods. Given that it should be mentioned what the general uncertainty range is and what the constraints are when using the data.

> Lacking an internationally agreed quantitative definition of cloud base heigh (CBH) and thus absolute reference values for ceilometer instruments, we are not able to present quantitative uncertainty values.
> In a report on the ceilometer intercomparison campaign CEILINEX2015, CBH differences of up to 70m were found between various ceilometer systems for liquid clouds, with even larger differences in precipitation conditions (Görsdorf et al., 2016). As there is no common definition of CBH, the instruments have been intercompared to each other to get an idea on the instrument-to-instrument variability, but uncertainties have not been presented.
> As there is no absolute reference, we consider the CBH in the presented ceilometer data set a best estimate for each respective sub-period. Constraints though are given for the calculation of trends: in this respect, the data should be treated as 3 incoherent datasets, that are generally too short to retrieve significant trends.
> An according statement has been added to the manuscript:

> >> Page 5, lines 31 pp.
> **As there is no absolute reference, we consider the CBH in the presented ceilometer data set a best estimate for each respective sub-period. Constraints though are given for the calculation of long-term trends: in this respect, the data should be treated as three incoherent datasets, each of them generally too short to retrieve significant trend information.**

> *Görsdorf, U. et al. (2016), The ceilometer inter-comparison campaign CEILINEX2015 – cloud detection and cloud base height. WMO Technical Conference on Meteorological and Environmental Instruments and Methods of Observation (CIMO TECO 2016), Madrid, Spain, 26 - 27 September 2016.*
> *https://www.wmo.int/pages/prog/www/IMOP/publications/IOM-125_TECO_2016/TECO_2016.html*

- The data set presented here consists of three time periods where the data was derived with different instruments. There is no overlap between all three instruments, which makes sense in terms of having one complete consistent data set. However, in case there was an overlap between the different instruments it would be interesting to add an comparison as an aspect referring to the sensitivity/uncertainty of the overall data set.

Unfortunately, the instruments have been operated without any overlap period.

- Looking at the link how to download the data it seams that the download has to be done on a monthly basis? Please provide one data file for the whole data set presented here or some way to easily download it as a whole since it is also presented as one data set in the manuscript. In case that is available but I was not able to find it, make sure that the website is easy to navigate.

Thank you for this suggestion. The webpage for data download now provides an additional link, containing a zip file of all data files.

- page 3, line 31: Could there be a microphysical reason for the disappearance of this cloud? Whas there some precipitation or graupel or similar observed?

Indeed, precipitation was observed during the occurrence of the low level cloud. Though we add this information to the manuscript, we will not discuss any microphysical implications since the reason for the cloud's disappearance cannot unambigously be identified.

>> now page 4, line 5 pp.
The cyclonic influence ends by about 15:30 UTC, when the clouds disappear **after a period with precipitation**, the long-wave radiation drops back to the 'clear' state, and air temperature decreases.

- page 5, line 6: How often did it happen that a month had more than 20% missing data and was excluded?

The months with technical problems leading to > 20 % missing data were 02/1993, 02-05/1997, 09/1999, 01-03/2000, 08/2000. This information has been added to the manuscript accordingly.

>> now page 5, line 11 pp.
The **apportioned** months July 1998 and August 2011 have been excluded, as well as all months that had more than 20% missing data **(February 1993, February to May 1997, July 1999, January to March 2000, August 2000, respectively)**.

- page 5, line 11: You say here that the signal could be masked by the different sensitivities of the different ceilometers, but you don't really discuss how large the uncertainty of the signal and the sensitivities of the different ceilometers are. It would be good to have some information here to relate to.

We understand that the wording was misleading. Rather than the 'different sensitivity' of the instruments, we intend to point out more general instrumental differences that affect the data retrieval. We have changed the sentence accordingly, pointing out the lack of a quantitative CBH definition, and the technical issues that lead to potential differences between the ceilometers.

>> now page 5, line 17 pp.
If a change in occurrence frequency of clouds over Ny-Ålesund occurred over the 25-year period, **it may still be masked by the effects of the diverse technical parameters of the different instruments (e.g. signal-to-noise ratio), or simply by the different applied algorithms for cloud determination. As the ceilometers have been sequentially operated without any overlap period, it is impossible to quantify the variability between the used instruments.**

- Figure 1: Mark which instrument is the ceilometer in the picture.

The ceilometer is now indicated by a white arrow.

>> Figure 1: The CL-51 ceilometer **(indicated by a white arrow)** located in the vicinity of the radiation measurements of the AWIPEV station at Ny-Ålesund, in April 2013.

- Figure 2: The abbreviations of the cloud types might need to be explained.

The explanation of the abbreviations has been added to the figure caption.

>> Figure 2: A frontal passage on 15 / 16 December 2016 in Ny-Ålesund. a: Schematic diagram of the warm front (red line) and cold front (blue line), their moving direction (black arrow), and associated clouds **(Ci = cirrus, Cc = cirrocumulus, Ac = altocumulus, Sc = stratocumulus, St = stratus)**, respectively.

- Figure 2: Change to a vector graphic.

Figure 2 will be submitted as vector graphic in the final version.

- Figure 5: The shading is very light and difficult to see. It disappeared on my print-out. Check again to make sure it can be clearly seen.

In the new version of Figure 5, we have added information on the longwave net radiation for the same periods, and also adjusted the shading.

- Table 1: In the text everywhere it is mentioned that the technology got better, the vertical resolution however went down from 1998-2011 to the actual data set from 2011 ongoing. Please comment this (and change in the text accordingly).

Although the vertical resolution of the actual instrument went down due to the longer laser pulse duration (100 ns), the overall performance is better compared to the older instrument due to the higher pulse energy (3 µJ vs. 1 µJ). Other technical advantages

are the lower power consumption and the possibility to retrieve backscatter profiles for the study of the boundary layer structure. The text has been changed accordingly, and more technical details have been added.

>> page 2, lines 27 pp.
Furthermore, it is likely that higher laser power and **improved** receiving hardware increased the sensitivity for cloud detection in the newer systems, potentially affecting the observed frequency of clear sky conditions. **Although the longer pulse duration of 75 ns for LD-40 compared to 100 ns for CL-51 has reduced the vertical resolution (Table 1), a higher laser pulse energy of 3 µJ instead of 1 µJ at the same pulse rate, respectively, surely increased the signal-to-noise ratio and thus the sensitivity for the detection of thin clouds.**

Small remarks,typos:
- page 1, line 22: exchange climate with global or add global to emphasize that this
  refers to a global mean.
- page 2, line 2: add , before which.
- page 2, line 26: replace was with were.
- page 3, line 9: avoid line break between number and unit.
- page 3, line 26: replace stably by stable.
- caption Fig. 3: replace symbols with dots.

The typos have been corrected in the revised manuscript version.

---

## Author Comment (AC3) · 11 Jul 2018

**General comment by authors:**

We appreciate the generally positive and encouraging feedback from the three reviewers. We also acknowledge the reviewer comments which led to improvements of the manuscript.

Below, the reviewer's comments are recalled in black, our response is given in blue, and changes to the manuscript are given green.

**Anonymous Referee #3**

The presented paper uses a 25 year data set of cloud base height measured with three different ceilometers (three different periods). The authors present a description of a case about a small cyclonic system based on CBH data and also longwave irradiance and temperature data. They also add ceilometer data in order to understand better the data provided by a cloud radar. Finally, a statistical analysis about the percentage of cloudy sky is discussed and also the time series of the CBH median, indicating that results could not be representative since the inhomogeneities in the time series caused by the changes on the instrument. In general this work fits with the scope of the journal and I consider it must be accepted after some minor revisions.

- Abstract: The abstract should not be equal to the introduction section, it should describe a little more what authors will do in the full paper.

> We have added a describing sentence to the abstract.

> >> page 1, lines 12 pp.
> **We explain the composition of the three sub-periods with different instrumentation contributing to the data set, and show examples of potential application areas.**

- The objective of the paper is not clear in the text.

> As any interpretation of data is out of the scope of ESSD, the article is intended as description pertaining to a data collection. By providing examples on potential application areas, we further underline the usefullness of the presented data set. We also discuss the quality of the data and point out their limitations. We have added an according statement to the Introduction:

> >> page 2, lines 12-13
> Here, we present a 25-year ceilometer cloud base height dataset from Ny-Ålesund, Svalbard, indicate the potential application areas **by providing several examples, and point out limitations of the data set with regard to trend analysis.**

- Figure 1 does not provide any significant information, it could be removed.

> Since the photo gives an impression of the environmental setting of the instrument and was appreciated by the other reviewers, we prefer to keep this figure.

- P4L6: This sentence should be in the introduction but not in this section.

> The sentence has been removed from section 3.2.

> >> **3.2 Cloud Base Height as Auxiliary for In-Situ and Remote Sensing Cloud Measurements**
> To approach the comprehensive characterization of macro- and micrpophysical cloud parameters in Ny-Ålesund, a 94 GHz frequency modulated continuous wave cloud radar (Küchler et al., 2017) has been installed…

- P4L7: The description of cloud radar fits better in the data section "data".

> Since the topic of this data paper is the ceilometer data set, we focus on the according ceilometer data description in the 'data section'. Section 3.2. should only provide an example for an application of the ceilometer data set. In this context, the description of the cloud radar data is of minor importance in terms of instrumental aspects, but the description is relevant to explain the necessity of simultaneous ceilometer measurements under certain atmospheric conditions. Therefore, we prefer to keep all cloud radar issues concentrated in one section.

- P5L18: The median of CBH trends should be quantified by Theil-Sen estimator or others, even homogeneity test could be done in order to stablish the inhomogeneities of the time series. In addition, the time series and trends of the percentage of cloudy sky should add more value to the paper.

- P5L28: but a homogeneity analysis could be done, or at least study the trends in the period of each ceilometer.

> Since the presented 25-year data set is actually a combination of data retrieved from 3 different instruments with different technical limitations and different retrieving algorithms, homogeneity is not given due to physical reasons. A homogeneity analysis therefor seems redundant.

> As the longest duration measured continuously with one instrument is 13 years, the time period for trend calculation is too short to retrieve any significant trend. This is a major issue we wanted to point out in our manuscript. As apparently it was not yet expressed in a sufficient manner, we added the following sentences:

> >> Page 5, lines 30 pp.
> **As there is no absolute reference, we consider the CBH in the presented ceilometer data set a best estimate for each respective sub-period. Constraints though are given for the calculation of long-term trends: in this respect, the data should be treated as three incoherent datasets, each of them generally too short to retrieve significant trend information.**

- P5L31-32: The results of this work do generally not prove that.

> We changed "prove" to "provide".

> >> Furthermore, the ceilometer data **provide** necessary auxiliary information for the retrieval of cloud parameters from the cloud radar.

- Figure 4: A legend could help to understand the bars.

Figure 4a already contained a legend. We have now added a similar legend to Figure 4b.